# Controllable Invariance
# through Adversarial Feature Learning

**Qizhe Xie, Zihang Dai, Yulun Du, Eduard Hovy, Graham Neubig**
Language Technologies Institute
Carnegie Mellon University
{qizhex, dzihang, yulund, hovy, gneubig}@cs.cmu.edu

## Abstract

Learning meaningful representations that maintain the content necessary for a particular task while filtering away detrimental variations is a problem of great interest in machine learning. In this paper, we tackle the problem of learning representations invariant to a *specific factor or trait* of data. The representation learning process is formulated as an adversarial minimax game. We analyze the optimal equilibrium of such a game and find that it amounts to maximizing the uncertainty of inferring the detrimental factor given the representation while maximizing the certainty of making task-specific predictions. On three benchmark tasks, namely fair and bias-free classification, language-independent generation, and lighting-independent image classification, we show that the proposed framework induces an invariant representation, and leads to better generalization evidenced by the improved performance.

## 1 Introduction

How to produce a data representation that maintains meaningful variations of data while eliminating noisy signals is a consistent theme of machine learning research. In the last few years, the dominant paradigm for finding such a representation has shifted from manual feature engineering based on specific domain knowledge to representation learning that is fully data-driven, and often powered by deep neural networks [Bengio et al., 2013]. Being universal function approximators [Gybenko, 1989], deep neural networks can easily uncover the complicated variations in data [Zhang et al., 2017], leading to powerful representations. However, how to systematically incorporate a desired invariance into the learned representation in a controllable way remains an open problem.

A possible avenue towards the solution is to devise a dedicated neural architecture that by construction has the desired invariance property. As a typical example, the parameter sharing scheme and pooling mechanism in modern deep convolutional neural networks (CNN) [LeCun et al., 1998] take advantage of the spatial structure of image processing problems, allowing them to induce more generic feature representations than fully connected networks. Since the invariance we care about can vary greatly across tasks, this approach requires us to design a new architecture each time a new invariance desideratum shows up, which is time-consuming and inflexible.

When our belief of invariance is specific to some attribute of the input data, an alternative approach is to build a probabilistic model with a random variable corresponding to the attribute, and explicitly reason about the invariance. For instance, the variational fair auto-encoder (VFAE) [Louizos et al., 2016] employs the maximum mean discrepancy (MMD) to eliminate the negative influence of specific "nuisance variables", such as removing the lighting conditions of images to predict the person's identity. Similarly, under the setting of domain adaptation, standard binary adversarial cost [Ganin and Lempitsky, 2015, Ganin et al., 2016] and central moment discrepancy (CMD) [Zellinger et al., 2017] have been utilized to learn features that are domain invariant. However, all these invariance

inducing criteria suffer from a similar drawback, which is they are defined to measure the divergence between a *pair* of distributions. Consequently, they can only express the invariance belief w.r.t. a pair of values of the random variable at a time. When the attribute is a multinomial variable that takes more than two values, combinatorial number of pairs (specifically, $O(n^2)$) have to be added to express the belief that the representation should be invariant to the attribute. The problem is even more dramatic when the attribute represents a structure that has exponentially many possible values (e.g. the parse tree of a sentence) or when the attribute is simply a continuous variable.

Motivated by the aforementioned drawbacks and difficulties, in this work, we consider the problem of learning a feature representation with the desired invariance. We aim at creating a unified framework that is (1) generic enough such that it can be easily plugged into different models, and (2) more flexible to express an invariance belief in quantities beyond discrete variables with limited value choices. Specifically, inspired by the recent advancement of adversarial learning [Goodfellow et al., 2014], we formulate the representation learning as a minimax game among three players: an *encoder* which maps the observed data deterministically into a feature space, a *discriminator* which looks at the representation and tries to identify a specific type of variation we hope to eliminate from the feature, and a *predictor* which makes use of the invariant representation to make predictions as in typical discriminative models. We provide theoretical analysis of the equilibrium condition of the minimax game, and give an intuitive interpretation. On three benchmark tasks from different domains, we show that the proposed approach not only improves upon vanilla discriminative approaches that do not encourage invariance, but also outperforms existing approaches that enforce invariant features.

## 2   Adversarial Invariant Feature Learning

In this section, we formulate our problem and then present the proposed framework of learning invariant features.

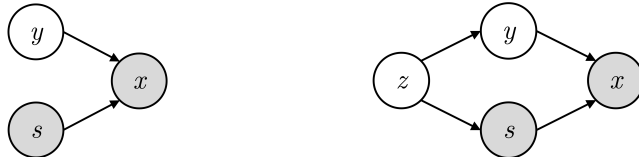

(a) $y$ and $s$ are marginally independent     (b) $y$ and $s$ are not marginally independent

Figure 1: Dependencies between $x, s, y$, where $x$ is the observation and $y$ is the target to be predicted. $s$ is the attribute to which the prediction should be invariant.

Given observation/input $x$, we are interested in the task of predicting the target $y$ based on the value of $x$ using a discriminative approach. In addition, we have access to some intrinsic attribute $s$ of $x$ as well as a prior belief that the prediction result should be invariant to $s$.

There are two possible dependency scenarios of $x, s$ and $y$ here: (1) $s$ and $y$ can be marginally independent. For example, in image classifications, lighting conditions $s$ and identities of persons $y$ are independent. The data generation process is $s \sim p(s), y \sim p(y), x \sim p(x \mid s, y)$. (2) In some cases, $s$ and $y$ are not marginally independent. For example, in fairness classifications, $s$ are the sensitive factors such as age and gender. $y$ can be the saving, credit and health condition of a person. $s$ and $y$ are related due to the inherent bias within the data. Using a latent variable $z$ to model the dependency between $s$ and $y$, the data generation process is $z \sim p(z), s \sim p(s \mid z), y \sim p(y \mid z), x \sim p(x \mid s, y)$. We show the corresponding dependency graphs in Figure 1.

Unlike vanilla discriminative models that outputs the conditional distribution $p(y \mid x)$, we model $p(y \mid x, s)$ to make predictions invariant to $s$. Our intuition is that, due to the explaining away effect, $y$ and $s$ are not independent when conditioned on $x$ although they can be marginally independent. Consequently, $p(y \mid x, s)$ is a more accurate estimation of $y$ than $p(y \mid x)$. Intuitively, this can inform and guide the model to remove information about undesired variations. For example, if we want to learn a representation of image $x$ that is invariant to the lighting condition $s$, the model can learn to "brighten" the input if it knows the original picture is dark, and vice versa. Also, in multi-lingual machine translation, a word with the same surface form may have different meanings in different languages. For instance, "gift" means "present" in English but means "poison" in German.

Hence knowing the language of a source sentence helps inferring the meaning of the sentence and conducting translation.

As the input $x$ can have highly complicated structure, we employ a dedicated model or algorithm to extract an expressive representation $h$ from $x$. Thus, when we extract the representation $h$ from $x$, we want the representation $h$ to preserve variations that are necessary to predict $y$ while eliminating information of $s$. To achieve the aforementioned goal, we employ a deterministic encoder $E$ to obtain the representation by encoding $x$ and $s$ into $h$, namely, $h = E(x, s)$. It should be noted here that we are using $s$ as an additional input. Given the obtained representation $h$, the target $y$ is predicted by a predictor $M$, which effectively models the distribution $q_M(y \mid h)$. By construction, instead of modeling $p(y \mid x)$ directly, the discriminative model we formulate captures the conditional distribution $p(y \mid x, s)$ with additional information coming from $s$.

Surely, feeding $s$ into the encoder by no means guarantees the induced feature $h$ will be invariant to $s$. Thus, in order to enforce the desired invariance and eliminate variations of factor $s$ from $h$, we set up an adversarial game by introducing a discriminator $D$ which inspects the representation $h$ and ensure that it is invariant to $s$. Concretely, the discriminator $D$ is trained to predict $s$ based on the encoded representation $h$, which effectively maximizes the likelihood $q_D(s \mid h)$. Simultaneously, the encoder fights to minimize the same likelihood of inferring the correct $s$ by the discriminator. Intuitively, the discriminator and the encoder form an adversarial game where the discriminator tries to detect an attribute of the data while the encoder learns to conceal it.

Note that under our framework, in theory, $s$ can be any type of data as long as it represents an attribute of $x$. For example, $s$ can be a real value scalar/vector, which may take many possible values, or a complex sub-structure such as the parse tree of a natural language sentence. But in this paper, we focus mainly on instances where $s$ is a discrete label with multiple choices. We plan to extend our framework to deal with continuous $s$ and structured $s$ in the future.

Formally, $E$, $M$ and $D$ jointly play the following minimax game:

$$\min_{E,M} \max_{D} J(E, M, D)$$

where

$$J(E, M, D) = \mathbb{E}_{x,s,y \sim p(x,s,y)} \left[ \gamma \log q_D(s \mid h = E(x, s)) - \log q_M(y \mid h = E(x, s)) \right] \quad (1)$$

where $\gamma$ is a hyper-parameter to adjust the strength of the invariant constraint, and $p(x, s, y)$ is the true underlying distribution that the empirical observations are drawn from.

Note that the problem of domain adaption can be seen as a special case of our problem, where $s$ is a Bernoulli variable representing the domain and the model only has access to the target $y$ when $s = $ "source domain" during training.

## 3 Theoretical Analysis

In this section, we theoretically analyze, given enough capacity and training time, whether such a minimax game will converge to an equilibrium where variations of $y$ are preserved and variations of $s$ are removed. The theoretical analysis is done in a non-parametric limit, i.e., we assume a model with infinite capacity. In addition, we discuss the equilibriums of the minimax game when $s$ is independent/dependent to $y$.

Since both the discriminator and the predictor only use $h$ which is transformed deterministically from $x$ and $s$, we can substitute $x$ with $h$ and define a joint distribution $\tilde{p}(h, s, y)$ of $h$, $s$ and $y$ as follows

$$\tilde{p}(h, s, y) = \int_x \tilde{p}(x, s, h, y)dx = \int_x p(x, s, y)p_E(h \mid x, s)dx = \int_x p(x, s, y)\delta(E(x, s) = h)dx$$

Here, we have used the fact that the encoder is a deterministic transformation and thus the distribution $p_E(h \mid x, s)$ is merely a delta function denoted by $\delta(\cdot)$. Intuitively, $h$ absorbs the randomness in $x$ and has an implicit distribution of its own. Also, note that the joint distribution $\tilde{p}(h, s, y)$ depends on the transformation defined by the encoder.

Thus, we can equivalently rewrite objective (1) as

$$J(E, M, D) = \mathbb{E}_{h,s,y \sim \tilde{p}(h,s,y)} \left[ \gamma \log q_D(s \mid h) - \log q_M(y \mid h) \right] \quad (2)$$

To analyze the equilibrium condition of the new objective (2), we first deduce the optimal discriminator $D$ and the optimal predictor $M$ for a given encoder $E$ and then prove the global optimality of the minimax game.

**Claim 1.** *Given a fixed encoder $E$, the optimal discriminator outputs $q_D^*(s \mid h) = \tilde{p}(s \mid h)$ and the optimal predictor corresponds to $q_M^*(y \mid h) = \tilde{p}(y \mid h)$.*

*Proof.* The proof uses the fact that the objective is functionally convex w.r.t. each distribution, and by taking the variations we can obtain the stationary point for $q_D$ and $q_M$ as a function of $\tilde{q}$. The detailed proof is included in the supplementary material A. $\qquad\square$

Note that the optimal $q_D^*(s \mid h)$ and $q_M^*(y \mid h)$ given in Claim 1 are both functions of the encoder $E$. Thus, by plugging $q_D^*$ and $q_M^*$ into the original minimax objective (2), it can be simplified as a minimization problem only w.r.t. the encoder $E$ with the following form:

$$\min_E J(E) = \min_E \mathop{\mathbb{E}}_{h,s,y \sim \tilde{q}(h,s,y)} \left[ \gamma \log \tilde{q}(s \mid h) - \log \tilde{q}(y \mid h) \right]$$
$$= \min_E -\gamma H(\tilde{q}(s \mid h)) + H(\tilde{q}(y \mid h)) \tag{3}$$

where $H(\tilde{q}(s \mid h))$ is the conditional entropy of the distribution $\tilde{q}(s \mid h)$.

**Equilibrium Analysis** As we can see, the objective (3) consists of two conditional entropies with different signs. Optimizing the first term amounts to maximizing the uncertainty of inferring $s$ based on $h$, which is essentially filtering out any information of $s$ from the representation. On the contrary, optimizing the second term leads to increasing the certainty of predicting $y$ based on $h$. Implicitly, the objective defines the equilibrium of the minimax game.

- **Win-win equilibrium:** Firstly, for cases where the attribute $s$ is entirely irrelevant to the prediction task (corresponding to the dependency graph shown in Figure 1a), the two terms can reach the optimum at the same time, leading to a win-win equilibrium. For example, with the lighting condition of an image removed, we can still/better classify the identity of the people in that image. With enough model capacity, the optimal equilibrium solution would be the same regardless of the value of $\gamma$.

- **Competing equilibrium:** However, there are cases where these two optimization objectives are competing. For example, in fair classifications, sensitive factors such as gender and age may help the overall prediction accuracies due to inherent biases within the data. In other words, knowing $s$ may help in predicting $y$ since $s$ and $y$ are not marginally independent (corresponding to the dependency graph shown in Figure 1b). Learning a fair/invariant representation is harmful to predictions. In this case, the optimality of these two entropies cannot be achieved simultaneously, and $\gamma$ defines the relative strengths of the two objectives in the final equilibrium.

## 4 Parametric Instantiation of the Proposed Framework

### 4.1 Models

To show the general applicability of our framework, we experiment on three different tasks including sentence generation, image classification and fair classifications. Due to the different natures of data of $x$ and $y$, here we present the specific model instantiations we use.

**Sentence Generation** We use multi-lingual machine translation as the testbed for sentence generation. Concretely, we have translation pairs between several source languages and a target language. $x$ is the source sentence to be translated and $s$ is a scalar denoting which source language $x$ belongs to. $y$ is the translated sentence for the target language.

Recall that $s$ is used as an input of $E$ to obtain a language-invariant representation. To make full use of $s$, we employ separate encoders $\mathrm{Enc}_s$ for sentences in each language $s$. In other words, $h = E(s, x) = \mathrm{Enc}_s(x)$ where each $\mathrm{Enc}_s$ is a different encoder. The representation of a sentence is captured by the hidden states of an LSTM encoder [Hochreiter and Schmidhuber, 1997] at each time step.

We employ a single LSTM predictor for different encoders. As often used in language generation, the probability $q_M$ output by the predictor is parametrized by an autoregressive process, i.e.,

$$q_M(y_{1:T} \mid h) = \prod_{t=1}^{T} q_M(y_t|y_{<t}, h)$$

where we use an LSTM with attention model [Bahdanau et al., 2015] to compute $q_M(y_t|y_{<t}, h)$.

The discriminator is also parameterized as an LSTM which gives it enough capacity to deal with input of multiple timesteps. $q_D(s \mid h)$ is instantiated with the multinomial distribution computed by a softmax layer on the last hidden state of the discriminator LSTM.

**Classification** For our classification experiments, the input is either a picture or a feature vector. All of the three players in the minimax game are constructed by feedforward neural networks. We feed $s$ to the encoder as an embedding vector.

## 4.2 Optimization

There are two possible approaches to optimize our framework in an adversarial setting. The first one is similar to the alternating approach used in Generative Adversarial Nets (GANs) [Goodfellow et al., 2014]. We can alternately train the two adversarial components while freezing the third one. This approach has more control in balancing the encoder and the discriminator, which effectively avoids saturation. Another method is to train all three components together with a gradient reversal layer [Ganin and Lempitsky, 2015]. In particular, the encoder admits gradients from both the discriminator and the predictor, with the gradient from the discriminator negated to push the encoder in the opposite direction desired by the discriminator. Chen et al. [2016b] found the second approach easier to optimize since the discriminator and the encoder are fully in sync being optimized altogether. Hence we adopt the latter approach. In all of our experiments, we use Adam [Kingma and Ba, 2014] with a learning rate of $0.001$.

# 5 Experiments

In this section, we perform empirical experiments to evaluate the effectiveness of proposed framework. We first introduce the tasks and corresponding datasets we consider. Then, we present the quantitative results showing the superior performance of our proposed framework, and discuss some qualitative analysis which verifies the learned representations have the desired invariance property.

## 5.1 Datasets

Our experiments include three tasks in different domains: (1) fair classification, in which predictions should be unaffected by nuisance factors; (2) language-independent generation which is conducted on the multi-lingual machine translation problem; (3) lighting-independent image classification.

**Fair Classification** For fair classification, we use three datasets to predict the savings, credit ratings and health conditions of individuals with variables such as gender or age specified as "nuisance variable" that we would like to not consider in our decisions [Zemel et al., 2013, Louizos et al., 2016]. The German dataset [Frank et al., 2010] is a small dataset with $1,000$ samples describing whether a person has a good credit rating. The sensitive nuisance variable to be factored out is gender. The Adult income dataset [Frank et al., 2010] has $45,222$ data points and the objective is to predict whether a person has savings of over $50,000$ dollars with the sensitive factor being age. The task of the health dataset[1] is to predict whether a person will spend any days in the hospital in the following year. The sensitive variable is also the age and the dataset contains $147,473$ entries. We follow the same 5-fold train/validation/test splits and feature preprocessing used in [Zemel et al., 2013, Louizos et al., 2016].

Both the encoder and the predictor are parameterized by single-layer neural networks. A three-layer neural network with batch normalization [Ioffe and Szegedy, 2015] is employed for the discriminator. We use a batch size of 16 and the number of hidden units is set to 64. $\gamma$ is set to 1 in our experiments.

**Multi-lingual Machine Translation**    For the multi-lingual machine translation task we use French to English (fr-en) and German to English (de-en) pairs from IWSLT 2015 dataset [Cettolo et al., 2012]. There are $198,435$ pairs of fr-en sentences and $188,661$ pairs of de-en sentences in the training set. In the test set, there are $4,632$ pairs of fr-en sentences and $7,054$ pairs of de-en sentences. We evaluate BLEU scores [Papineni et al., 2002] using the standard Moses `multi-bleu.perl` script. Here, $s$ indicates the language of the source sentence.

We use the OpenNMT [Klein et al., 2017] in our multi-lingual MT experiments[2]. The encoder is a two-layer bidirectional LSTM with 256 units for each direction. The discriminator is a one-layer single-directional LSTM with 256 units. The predictor is a two-layer LSTM with 512 units and attention mechanism [Bahdanau et al., 2015]. We follow Johnson et al. [2016] and use Byte Pair Encoding (BPE) subword units [Sennrich et al., 2016] as the cross-lingual input. Every model is run for 20 epochs. $\gamma$ is set to $8$ and the batch size is set to $64$.

**Image Classification**    We use the Extended Yale B dataset [Georghiades et al., 2001] for our image classification task. It comprises face images of 38 people under 5 different lighting conditions: upper right, lower right, lower left, upper left, or the front. The variable $s$ to be purged is the lighting condition. The label $y$ is the identity of the person. We follow Li et al. [2014], Louizos et al. [2016]'s train/test split and no validation is used: $38 \times 5 = 190$ samples are used for training and all other $1,096$ data points are used for testing.

We use a one-layer neural network for the encoder and a one-layer neural network for prediction. $\gamma$ is set to 2. The discriminator is a two-layer neural network with batch normalization. The batch size is set to 16 and the hidden size is set to $100$.

## 5.2  Results

**Fair Classification**    The results on three fairness tasks are shown in Figure 2. We compare our model with two prior works on learning fair representations: Learning Fair Representations (LFR) [Zemel et al., 2013] and Variational Fair Autoencoder (VFAE) [Louizos et al., 2016]. Results of VAE and directly using $x$ as the representation are also shown.

We first study how much information about $s$ is retained in the learned representation $h$ by using a logistic regression to predict factor $s$. In the top row, we see that $s$ cannot be recognized from the representations learned by three models targeting at fair representations. The accuracy of classifying $s$ is similar to the trivial baseline predicting the majority label shown by the black line.

The performance on predicting label $y$ is shown in the second row. We see that LFR and VFAE suffer on Adult and German datasets after removing information of $s$. In comparison, our model's performance does not suffer even when making fair predictions. Specifically, on German, our model's accuracy is $0.744$ compared to $0.727$ and $0.723$ achieved by VFAE and LFR. On Adult, our model's accuracy is $0.844$ while VFAE and LFR have accuracies of $0.813$ and $0.823$ respectively. On the health dataset, all models' performances are barely better than the majority baseline. The unsatisfactory performances of all models may be due to the extreme imbalance of the dataset, in which $85\%$ of the data has the same label.

We also investigate how fair representations would alleviate biases of machine learning models. We measure the unbiasedness by evaluating models' performances on identifying minority groups. For instance, suppose the task is to predict savings with the nuisance factor being age, with savings above a threshold of $\$50,000$ being adequate, otherwise being insufficient. If people of advanced age generally have fewer savings, then a biased model would tend to predict insufficient savings for those with an advanced age. In contrast, an unbiased model can better factor out age information and recognize people that do not fit into these stereotypes.

Concretely, for groups pooled by each possible value of $y$, we seek for the minority $s$ in each of these groups and define the minority $s$ as the biased category for the group. Then we first calculate the accuracy on each biased category and report the average performance for all categories. We do not compute the instance-level average performance since one category may hold the dominant amount of data among all categories.

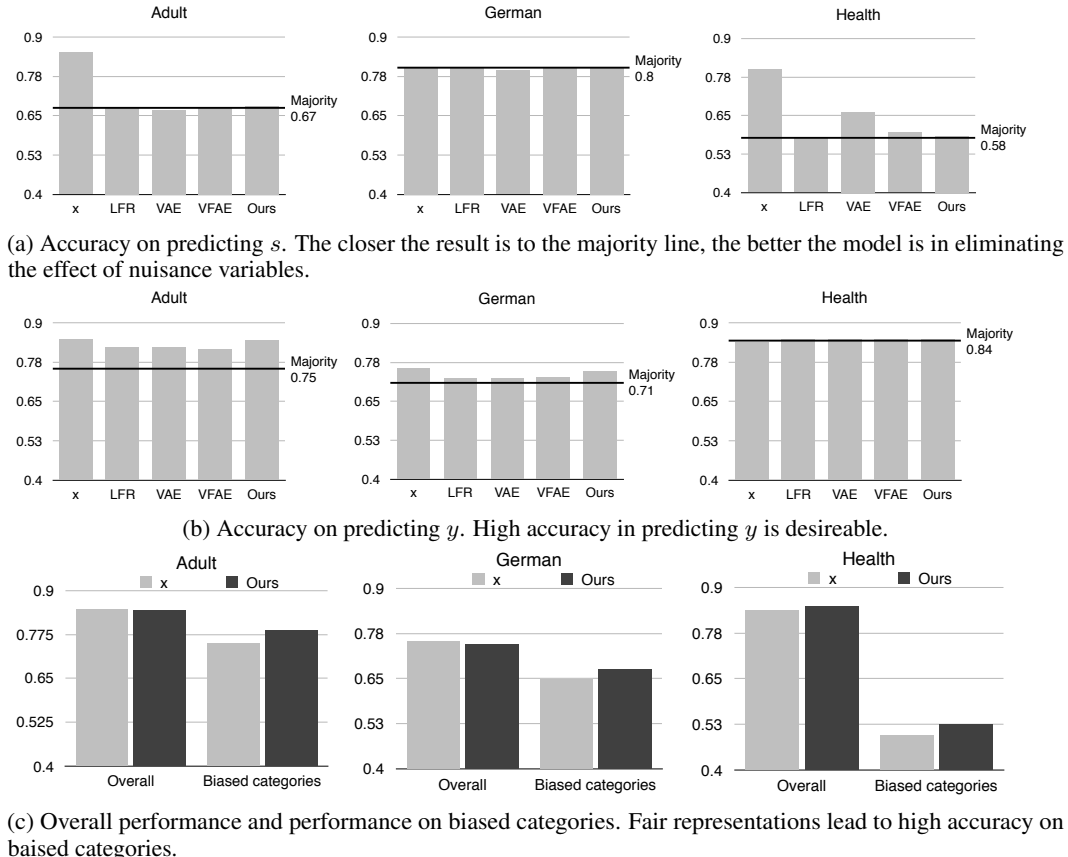

(a) Accuracy on predicting $s$. The closer the result is to the majority line, the better the model is in eliminating the effect of nuisance variables.

(b) Accuracy on predicting $y$. High accuracy in predicting $y$ is desireable.

(c) Overall performance and performance on biased categories. Fair representations lead to high accuracy on baised categories.

Figure 2: Fair classification results on different representations. $x$ denotes directly using the observation $x$ as the representation. The black lines in the first and the second row show the performance of predicting the majority label. "Biased categories" in the third row are explained in the fourth paragraph of Section 5.2.

| Model | test (fr-en) | test (de-en) |
|---|---|---|
| Bilingual Enc-Dec [Bahdanau et al., 2015] | 35.2 | 27.3 |
| Multi-lingual Enc-Dec [Johnson et al., 2016] | 35.5 | 27.7 |
| Our model | **36.1** | **28.1** |
|    w.o. discriminator | 35.3 | 27.6 |
|    w.o. separate encoders | 35.4 | 27.7 |

Table 1: Results on multi-lingual machine translation.

As shown in the third row of Figure 2, on German and Adult, we achieve higher accuracy on the biased categories, even though our overall accuracy is similar to or lower than the baseline which does not employ fairness constraints. Specifically, on Adult, our performance on the biased categories is 0.788 while the baseline's accuracy is 0.748. On German, our accuracy on biased categories is 0.676 while the baseline achieves 0.648. The results show that our model is able to learn a more unbiased representation.

**Multi-lingual Machine Translation**    The results of systems on multi-lingual machine translation are shown in Table 1. We compare our model with attention based encoder-decoder trained on bilingual data [Bahdanau et al., 2015] and multi-lingual data [Johnson et al., 2016]. The encoder-decoder trained on multi-lingual data employs a single encoder for both source languages. Firstly, both multi-lingual systems outperform the bilingual encoder-decoder even though multi-lingual systems use similar number of parameters to translate two languages, which shows that learning

| Method | Accuracy of classifying $s$ | Accuracy of classifying $y$ |
|---|---|---|
| Logistic regression | 0.96 | 0.78 |
| NN + MMD [Li et al., 2014] | - | 0.82 |
| VFAE [Louizos et al., 2016] | **0.57** | 0.85 |
| Ours | **0.57** | **0.89** |

Table 2: Results on Extended Yale B dataset. A better representation has lower accuracy of classifying factor $s$ and higher accuracy of classifying label $y$

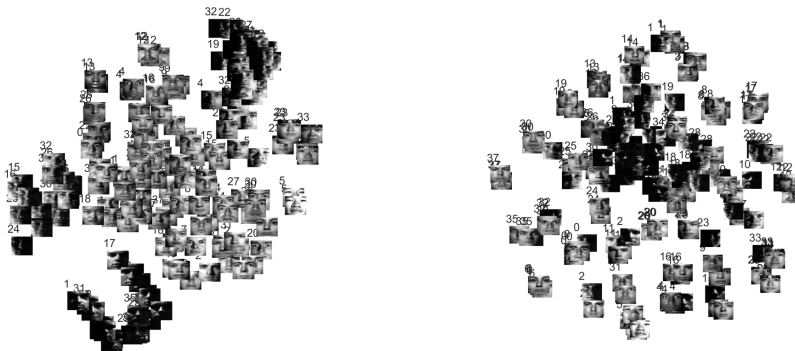

(a) Using the original image $x$ as the representation      (b) Representation learned by our model

Figure 3: t-SNE visualizations of images in the Extended Yale B. The original pictures are clustered by the lighting conditions, while the representation learned by our model is clustered by identities of individuals

invariant representation leads to better generalization in this case. The better generalization may be due to transferring statistical strength between data in two languages.

Comparing two multi-lingual systems, our model outperforms the baseline multi-lingual system on both languages, where the improvement on French-to-English is $0.6$ BLEU score. We also verify the design decisions in our framework by ablation studies. Firstly, without the discriminator, the model's performance is worse than the standard multi-lingual system, which rules out the possibility that the gain of our model comes from more parameters of separating encoders. Secondly, when we do not employ separate encoders, the model's performance deteriorates and it is more difficult to learn a cross-lingual representation, which

- verifies the theoretical advantage of modeling $p(y \mid x, s)$ instead of $p(y \mid x)$ as mentioned in Section 2. Intuitively, German and French have different grammars and vocabulary, so it is hard to obtain a unified semantic representation by performing the same operations.

- means that the encoder needs to have enough capacity to reach the equilibrium in the minimax game. We also observe that the discriminator needs enough capacity to provide faithful gradients towards the equilibrium. Specifically, instantiating the discriminator with feedforward neural network w./w.o. attention mechanism [Bahdanau et al., 2015] does not work in our experiments.

**Image Classification**   We report the results in Table 2 with two baselines [Li et al., 2014, Louizos et al., 2016] that use MMD regularizations to remove lighting conditions. The advantage of factoring out lighting conditions is shown by the improved accuracy $89\%$ for classifying identities, while the best baseline achieves an accuracy of $85\%$.

In terms of removing $s$, our framework can filter the lighting conditions since the accuracy of classifying $s$ drops from $0.96$ to $0.57$, as shown in Table 2. We also visualize the learned representation by t-SNE [Maaten and Hinton, 2008] in comparison to the visualization of original pictures in Figure 3. We see that, without removing lighting conditions, the images are clustered based on the lighting conditions. After removing information of lighting conditions, images are clustered according to the identity of each person.

# 6   Related Work

As a specific case of our problem where $s$ takes two values, domain adaption has attracted a large amount of research interest. Domain adaptation aims to learn domain-invariant representations that are transferable to other domains. For example, in image classification, adversarial training has been shown to able to learn an invariant representation across domains [Ganin and Lempitsky, 2015, Ganin et al., 2016, Bousmalis et al., 2016, Tzeng et al., 2017] and enables classifiers trained on the source domain to be applicable to the target domain. Moment discrepancy regularizations can also effectively remove domain specific information [Zellinger et al., 2017, Bousmalis et al., 2016] for the same purpose. By learning language-invariant representations, classifiers trained on the source language can be applied to the target language [Chen et al., 2016b, Xu and Yang, 2017].

Works targeting the development of fair, bias-free classifiers also aim to learn representations invariant to "nuisance variables" that could induce bias and hence makes the predictions fair, as data-driven models trained using historical data easily inherit the bias exhibited in the data. Zemel et al. [2013] proposes to regularize the $\ell_1$ distance between representation distributions for data with different nuisance variables to enforce fairness. The Variational Fair Autoencoder [Louizos et al., 2016] targets the problem with a Variational Autoencoder [Kingma and Welling, 2014, Rezende et al., 2014] approach with maximum mean discrepancy regularization.

Our work is also related to learning disentangled representations, where the aim is to separate different influencing factors of the input data into different parts of the representation. Ideally, each part of the learned representation can be marginally independent to the other. An early work by Tenenbaum and Freeman [1997] propose a bilinear model to learn a representation with the style and content disentangled. From information theory perspective, Chen et al. [2016a] augments standard generative adversarial networks with an inference network, whose objective is to infer part of the latent code that leads to the generated sample. This way, the information carried by the chosen part of the latent code can be retained in the generative sample, leading to disentangled representation.

As we have discussed in Section 1, these methods bear the same drawback that the cost used to regularize the representation is pairwise, which does not scale well as the number of values that the attribute can take could be large. Louppe et al. [2016] propose an adversarial training framework to learn representations independent to a categorical or continuous variable. A basic assumption in their theoretical analysis is that the attribute is irrelevant to the prediction, which limits its capabilities in analyzing the fairness classifications.

# 7   Conclusion

In sum, we propose a generic framework to learn representations invariant to a specified factor or trait. We cast the representation learning problem as an adversarial game among an encoder, a discriminator, and a predictor. We theoretically analyze the optimal equilibrium of the minimax game and evaluate the performance of our framework on three tasks from different domains empirically. We show that an invariant representation is learned, resulting in better generalization and improvements on the three tasks.

## Acknowledgement

We thank Shi Feng, Di Wang and Zhilin Yang for insightful discussions. This research was supported in part by DARPA grant FA8750-12-2-0342 funded under the DEFT program.

## Footnotes

[1] www.heritagehealthprize.com

[2]Our MT code is available at https://github.com/qizhex/Controllable-Invariance

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

# A   Supplementary Material: Proofs

The proof for Claim 1:

**Claim.** *Given a fixed encoder $E$, the optimal discriminator outputs $q_D^*(s \mid h) = \tilde{p}(s \mid h)$. The optimal predictor corresponds to $q_M^*(y \mid h) = \tilde{p}(y \mid h)$.*

*Proof.* We first prove the optimal solution of the discriminator. With a fixed encoder, we have the following optimization problem

$$\min_{q_D} \quad -J(E, M, D)$$

$$\text{s.t.} \quad \sum_s q_D(s \mid h) = 1, \forall h$$

Then $L = J(E, M, D) - \sum_h \lambda(h)(\sum_s q_D(s \mid h) - 1)$ is the Lagrangian dual function of the above optimization problem where $\lambda(h)$ are the dual variables introduced for equality constraints.

The optimal $D$ satisfies the following equation

$$0 = \frac{\partial L}{\partial q_D^*(s \mid h)}$$

$$\iff \quad 0 = -\frac{\partial J}{\partial q_D^*(s \mid h)} - \lambda(h) \tag{4}$$

$$\iff \quad \lambda(h) = -\frac{\sum_y \tilde{q}(h, s, y)}{q_D^*(s \mid h)}$$

$$\iff \quad \lambda(h) q_D^*(s \mid h) = -\tilde{q}(s, h)$$

Summing w.r.t. $s$ on both sides of the last line of Eqn. (4) and using the fact that $\sum_s q_D^*(s \mid h) = 1$, we get

$$\lambda(h) = -\tilde{q}(h) \tag{5}$$

Substituting Eqn. 5 back into Eqn. 4, we can prove the optimal discriminator is

$$q_D^*(s \mid h) = \tilde{q}(s \mid h)$$

Similarly, taking derivation w.r.t. $q_M(y \mid h)$ and setting it to 0, we can prove $q_M^*(y \mid h) = \tilde{q}(y \mid h)$. $\qquad \square$

