[Reviews · NeurIPS 2017]

Reviewer 1



Summary: This paper is about learning representations designed with specific invariances. The authors propose a formulation that builds on Adversarial Training (Goodfellow et al 2014) by adding a new component to that handles invariances. The new model they proposed includes, an encoder, a discriminator and a predictor since their approach focuses on the classification results. They evaluate their approach on three benchmarks and show results that compete with the state of the art. Positive Points: -The paper is well written and easy to follow -The formulation is interesting and the claims are sound, including the invariance learning in the hidden representation and formulating it using conditional entropies is novel. -The evaluation outperforms the state of the art in some cases. Negative Points: -The results on the Fair Classification as well as the Multi-lingual Machine Translation are relatively close to the state-of-the-art which makes me question the benefit of using their approach versus a VAE. -The authors should explain why the VAE outperformed there approach in the case of "Health" data in Figure1(a). More comments on the results would be useful. I would encourage the authors to evaluate their approach on CELEB-A dataset and show visualization comparing to GANs and VAEs since CELEB-A includes many invariances that could be learned and tested on.

Reviewer 2



This paper tackles the problem of learning fair representations, that is, deducing the representation invariant to "nuisance" variables, and proposes adversarial training between encoder, predictor, and discriminator such that, given representation from the encoder, discriminator tries to predict nuisance variables and the predictor aims to predict the label. The idea is interesting, and the presented theoretical analysis is nice. Also demonstration with three tasks looks to show its effectiveness as it outperforms baselines. But I concern that, as in the multi-lingual machine translation task, using invariant representation would improve the performance of original tasks. In the 1st Fair Classifications task, it did not improve the performance with original input x, and in the 3rd Image Classification task, logistic regression with x seems a too weak baseline. Writing is mostly clear and easy to follow. Detailed comments: - In the equation below the line 100, it was not clear to me how to deduce 3rd term from 2nd term. - In Fair Classifications task, single hidden layer NN seems too shallow for encoder and decoder, and have you tried deeper NNs for them?

Reviewer 3



The paper proposes to learn invariant features using adversarial training. Given s a nuisance factor (s attribute of x), a discriminator tries to predict the nuisance factors s (an attribute of the input) given a encoder representation h=E(x,s) , and an encode rE tries to minimize the prediction of nuisance factor and and to predict the desired output. The encoder is function of x and s. Novelty: The paper draws some similarity with Ganian et al on unsupervised domain adaptation and their JMLR version. The applications to Multilingual machine translation and fairness applications are to the best of the knowledge of the reviewer new in this context and are interesting. Comments: - The application of the method to invariant features learning in object recognition is not realistic. In machine translation s being the one hot of the language at hand or in fairness s being an attribute of the input , is realistic. But if one wants invariance to rotation or lightning one can not supply the angle or the lightening to the encoder. E(x,s) is not realistic for geometric invariance i.e when s is indeed hidden and not given at test time. Have you tried in this case for instance E being just E(x)? See for instance this work https://arxiv.org/pdf/1612.01928.pdf that uses actually a very similar training with just E(x) - I think the paper should be more focused on the multiple sources / single target domain adaptation aspect. Geometric invariances are confusing in this setting, since the nuisance factor can not be assumed known at test time. - How would you extend this to deal with continuous nuisances factors s?